# Copper Adsorption on Lignin for the Removal of Hydrogen Sulfide

**DOI:** 10.3390/molecules25235577

**Published:** 2020-11-27

**Authors:** Miroslav Nikolic, Marleny Cáceres Najarro, Ib Johannsen, Joseph Iruthayaraj, Marcel Ceccato, Anders Feilberg

**Affiliations:** 1Department of Engineering, Aarhus University, Hangøvej 2, 8200 Aarhus, Denmark; nmiroslav@ymail.com (M.N.); ibj@eng.au.dk (I.J.); josedeva76@gmail.com (J.I.); mceccato@inano.au.dk (M.C.); 2Interdisciplinary Nanoscience Center (iNANO), Aarhus University, Gustav Wieds Vej 14, 8000 Aarhus, Denmark

**Keywords:** kraft lignin, adsorbent material, biobased materials, copper adsorption, H_2_S adsorption, H_2_S removal

## Abstract

Lignin is currently an underutilized part of biomass; thus, further research into lignin could benefit both scientific and commercial endeavors. The present study investigated the potential of kraft lignin as a support material for the removal of hydrogen sulfide (H_2_S) from gaseous streams, such as biogas. The removal of H_2_S was enabled by copper ions that were previously adsorbed on kraft lignin. Copper adsorption was based on two different strategies: either directly on lignin particles or by precipitating lignin from a solution in the presence of copper. The H_2_S concentration after the adsorption column was studied using proton-transfer-reaction mass spectrometry, while the mechanisms involved in the H_2_S adsorption were studied with X-ray photoelectron spectroscopy. It was determined that elemental sulfur was obtained during the H_2_S adsorption in the presence of kraft lignin and the differences relative to the adsorption on porous silica as a control are discussed. For kraft lignin, only a relatively low removal capacity of 2 mg of H_2_S per gram was identified, but certain possibilities to increase the removal capacity are discussed.

## 1. Introduction

In recent times, significant efforts are being placed into the substitution of fossil-based natural gas, also known as synthetic natural gas (SNG), with renewable biomethane (CH_4_), which is mainly obtained from upgrading biogas via the removal of CO_2_, H_2_S, and other trace gases [1,2]. Before upgrading, the biogas composition can contain between 80 and 4000 ppm of H_2_S, depending on the feedstock material [3]. The corrosive and harmful nature of H_2_S [3,4] requires it to be efficiently removed for further uses of biogas/biomethane. Numerous technologies already exist for the removal of H_2_S from gaseous streams, including solid reagents (oxides and salts), activated carbon, biological removal, solvent absorption, and chelating solutions; however, all of them have various deficiencies, such as incomplete absorption, methane absorption, operational difficulties, or high costs involved [5]. Therefore, novel approaches are being explored in order to uncover more effective solutions.

Lignin, the second most abundant biopolymer after cellulose, is currently an underutilized component of lignocellulosic biomass due to its heterogeneity, recalcitrant nature, and difficulties in processing, amongst others [6,7,8]. Lignin is often treated as waste material and the largest quantities are currently being burnt for energy as a low-value fuel [9]. With the emergence of biorefineries, the availability of lignin is only expected to increase in the future, making the valorization of lignin to various chemicals and materials of high importance [10].

In material applications, lignin has been studied for many years as a precursor for activated carbon adsorbents [11,12]. Activated carbons with structured pores and large internal surface areas have high adsorption capacities toward various substances [13], including H_2_S [14]. The high cost of activated carbons [15], together with increased environmental considerations [16], has influenced research to move toward lower-cost natural adsorbent materials. Lignin has also been studied for its natural adsorbent properties, without the utilization of carbonization processes, but mainly for adsorbing metal ions for use in, e.g., water purification [17]. The adsorption of metal ions on lignin is mostly driven by a large fraction of functional groups that are available due to various processes of lignin isolation from nature [17]. For example, milled wood lignin, which is a reasonable representation of native lignin, contains mostly aliphatic alcohol groups and only around 1 mmol/g of phenolic OH and around 0.2 mmol/g of carboxyl groups [18]. On the other hand, the most abundant technical lignins, such as kraft lignin and lignosulfonates, differ from that. Kraft process leads to around 3 mmol/g of phenolic OH and around 0.5 mmol/g of carboxyl groups [18,19], while lignosulfonates are typically characterized by 5–8 wt% of sulfur all in the form of a low pKa sulfonic groups, which can represent 13% of lignosulfonates’ weight [20]. Since transition metals are also effective in the removal of H_2_S [3,5], we have chosen to premiere a study of utilizing non-carbonized lignin as a support for copper ions in the removal of H_2_S. 

Ferric oxide and chelated ferric solutions are the most often used among the H_2_S removal technologies centered on transition metals [5,14] due to the high removal capacity and the more benign environmental effects of iron compared to most of the other transition metals. However, we have opted for copper in our study for the following reasons. First, a recent study has shown the good removal capacity of H_2_S by a copper (II) solution due to the precipitation of copper (II) sulfide [21]. When utilized heterogeneously (as adsorbed on other solid supports), copper has shown better results compared to several metal ions, including iron [22]. In addition, if the use of lignin in water purification is commercialized in the future, large amounts of lignin saturated with heavy metal ions, such as copper, can easily become available. To the best of our knowledge, the application of lignin as a support material for copper in gas purification has not been reported previously. The present study focused on establishing copper adsorption mechanisms on a lignin structure and determined their removal capacity toward H_2_S. The results offer an initial insight into potentially extending the value chain of lignin into biogas cleaning, going beyond water purification. The system was tested for the treatment of a model system consisting of H_2_S in N_2_ in order to characterize the process under controlled conditions. H_2_S is the dominant sulfur compound in many applications, including biogas [23,24,25], and its removal will not be significantly influenced by the presence of small amounts of other sulfur compounds. A lignin-Cu adsorbent would typically be applied as a final purification step to protect, e.g., catalysts from sulfur poisoning [24,25]. It is therefore relevant to test a model system containing only H_2_S.

## 2. Materials and Methods

Softwood kraft lignin BioPiva 100 was purchased from UPM (Helsinki, Finland). Copper (II) bromide and tetrahydrofurane (THF) were purchased from Sigma Aldrich (Vandtårnsvej, Denmark) and used as obtained. Davison silica gel grade 12 60/80 mesh was sourced from Supelco Analytical (Bellefonte, PA, USA). A hydrogen sulfide (H_2_S) gas cylinder (720 ppm) with nitrogen as a carrier gas was procured from AGA (Fredericia, Denmark).

### 2.1. Copper Adsorption

Before further treatment, the kraft lignin was washed three times with Milli-Q water (Millipore, Denmark) and dried at 45 °C. This process allowed for removing soluble compounds, e.g., sulfate ions [26]. In one set of experiments, 2 g of washed kraft lignin was dispersed in a 0.15 M solution of CuBr_2_ and stirred overnight. The following day, lignin was separated via centrifugation, washed several times, and dried at 45 °C before further use. 

In the second set of experiments, washed kraft lignin was dissolved in THF, mixed with a CuBr_2_ water solution, and stirred overnight. The copper concentration (0.15 M) was kept the same as in the previous experiment, while a particular THF:water ratio of 5:2 by weight was selected based on trial and error to avoid precipitation. The following day, the THF was removed using a rotary evaporator and the lignin particles were washed with water several times and dried at 45 °C before further use. 

Thermogravimetric analysis (TGA) was used to establish the amount of copper adsorbed on solid lignin. Kraft lignin powder before and after adsorption was heated from room temperature to 800 °C at a rate of 10 °C/min under nitrogen.

### 2.2. H_2_S Removal Experiments

For the H_2_S removal experiments, lignin was mixed with Davison porous silica in a 1:3 weight ratio and the adsorption experiments were repeated in the same manner. The H_2_S removal experiments were performed in a laboratory-developed test at room temperature to determine the capacity of lignin (with adsorbed copper) for H_2_S removal. Controls utilizing only silica with adsorbed copper, silica without adsorbed copper, and a silica/lignin mixture without adsorbed copper were tested to enable normalization and to calculate the adsorption capacity of lignin. 

The material was packed in a Teflon column (length 90 mm, inner diameter 3.2 mm, 0.5 g of adsorbent material) by placing glass wool on both ends of the column to prevent material loss during the experiments. The H_2_S was diluted with compressed air and the flow rates were controlled with gas mass flow controllers (Bronkhorst EL FLOW, Ruurlo, The Netherlands). Air containing 5 ppm of H_2_S (the H_2_S concentration was chosen after initial screening tests) was passed through a Teflon column at 300 mL/min. To determine the influence of moisture on the H_2_S removal, the experiments were performed utilizing non-conditioned compressed air (14% relative humidity (RH)) and air passed through a water impinger to obtain 90–100% relative humidity. 

A proton-transfer-reaction mass spectrometer (PTR-MS, Ionicon Analytik GmbH, Innsbruck, Austria), which allowed for online data collection, was used for monitoring the H_2_S (mass-to-charge ratio (*m*/*z*) of 35) removal by measuring after the adsorption column. The instrument used was a high-sensitivity quadrupole PTR-MS. The PTR-MS was operated under standard drift tube conditions with a voltage at 600 V and pressure close to 2.2 mbar, while the inlet and chamber temperature was set to 75 °C.

The experiments were stopped when a 10% (0.5 ppm of H_2_S) breakthrough was reached. An empty column was run before the actual experiment in order to establish the maximum measured level of H_2_S with PTR-MS under the set conditions. Although silica gel is hydrophilic, the humidity did not affect any of the packed columns over the course of the experiments to any significant degree. The H_2_S signal was therefore not corrected for humidity dependency in order to determine percentage removal efficiency [27]. The removal capacity was calculated based on the inlet and outlet H_2_S concentrations, flow rate, breakthrough time, and the volume of adsorbent material.

### 2.3. X-ray Photoelectron Spectroscopy (XPS)

The XPS analyses were performed with a Kratos Axis UltraDLD spectrometer (Kratos Analytical Ltd., Manchester, UK) using a monochromated Al X-ray source (1486.7 eV) operated at 150 W and pressure in the main chamber in the 10^−9^ torr range. The information collected was related to approximately the top 10 nm of the sample surface. Survey scans were performed with a 1.0 eV step size and a 160 eV analyzer pass energy, while the high-resolution scans were recorded at a 20 eV pass energy and a 0.1 eV step size. The spectra were processed using CasaXPS software (Version 2.3.20, Casa Software Ltd., Terrace Teignmouth, UK) and calibrated by setting the main C1s peak to 284.8 eV. The atomic surface concentration (atom%) was determined from the survey spectra and represents the average of the measurements taken at two different spots on each sample.

## 3. Results and Discussion

### 3.1. Copper Adsorption

The formation of stable surface complexes between the Cu(II) cations and the lignin substrate has been documented [28]. The surface of lignin particles is rich in functional groups, such as carboxyl and alcohol groups, that have a high affinity toward metal ions. The softwood kraft lignin used in this study was previously characterized to have 0.5 mmol/g of carboxyl groups, 2.0 mmol/g of aliphatic OH, and 4.8 mmol/g of various phenolic OH species [29]. Past research has shown higher adsorption of copper ions on lignin at higher pH levels in general [30,31], suggesting the importance of deprotonating surface carboxyl groups for obtaining a high adsorption capacity. At the same time, for certain metal ions, such as Fe(III), coordination with phenolic oxygen was established [32] and should not be neglected. Figure 1 shows the thermogravimetric analysis for kraft lignin and the samples with adsorbed copper. TGA is a technique that is broadly used to study the adsorption of materials [33,34,35,36], and in the present study, it allowed for establishing the amount of copper adsorbed on powder lignin. Kraft lignin had a residual weight of around 40% at 800 °C, which is in line with other research results for softwood kraft lignin with the same applied heating rate [37]. As expected, samples with adsorbed copper had a higher residual weight at 800 °C compared to pure kraft lignin. The copper adsorption capacity of lignin was calculated based on the TGA curves as the difference between the residual weight at 800 °C for lignin before and after adsorption. When Cu(II) was adsorbed on lignin particles, a capacity of 14 ± 5 mg/g of lignin was determined, in agreement with previous results for various lignin particles [30,31,38,39,40].

On the other hand, by dissolving lignin in the THF/H_2_O mixture and later precipitating it, the capacity for copper adsorption increased several times to 72 ± 13 mg/g of lignin. By solubilizing lignin, thus disentangling the polymer structure and solvating all the functional groups [41], the number of functional groups accessible to copper ions increased compared to the particle form, thereby increasing the total number of adsorption sites for copper ions. This effect can be considered as a general concept regarding the accessibility of functional groups. For example, Sun et al. [42] have shown with the use of qNMR combined with wet chemistry that the accessible amine content is lower on modified silica particles compared to the amine content of the same dissolved silica. It is well known that in solution, Cu(II) can form complexes of various stabilities with amines, acids, and hydroxy-acids [43]. For hydroxy-acids, such as dihydroxybenzoic acid, it has been shown that Cu complexes can remain stable without being dissolved through a combination of electrostatic interactions and an extended network of hydrogen bonds due to retention of water molecules [44]. Similar mechanisms are likely involved for macromolecular lignin-Cu complexes; thus, once copper is complexed in solution, it can remain stable upon precipitation. Considering all these, it is not surprising that the adsorption capacity increased for the same lignin. Metal ion adsorption on lignin is most often exploited for water purification, in which case, adsorption on already-formed particles is the most logical route [17]. However, this result shows that for other applications, alternative approaches should also be considered. 

### 3.2. H_2_S Removal Experiments

The performances of the materials for H_2_S adsorption were characterized in terms of the breakthrough capacity using the method described in Section 2.2. The utilized PTR-MS instrument allowed for continuous and direct measurement of the H_2_S levels with a time resolution of 100 ms. Figure 2 shows an example of breakthrough curves during screening tests for a system with and without adsorbed copper. It was clear from the tested controls that without copper, breakthroughs were occurring practically immediately and that copper was responsible for the H_2_S removal from the air stream. Initially, tests were done by filling the column with only lignin. However, due to the kraft lignin powder packing density and low porosity of the particles, it was not possible to obtain stable airflow through the column (because of the large pressure build-up and likely occurrence of air channeling). The typical surface area of kraft lignin powder is in the range of 0.1–30 m^2^/g [45,46,47], which is low for powder adsorbents, and for this reason, Davison porous silica gel with a surface area of 750 m^2^/g (data obtained from the supplier) was introduced. The selected 3:1 weight ratio of silica powder to lignin powder provided a distribution of particles in the column such that there was no airflow fluctuation through the column during the experiment. Tests were also made with only porous silica as the adsorbent material (with and without copper) to enable the normalization of the results for lignin/silica mixtures.

Table 1 shows the determined breakthrough capacities for H_2_S removal. It can be seen that the higher removal capacity was obtained for the system that could adsorb more copper since the native materials (silica and kraft lignin) did not show tendencies toward adsorbing H_2_S at significant levels (data not shown). Past research has shown that the levels of copper adsorption by non-modified amorphous silica are around 5 mg/g [48,49,50], which is lower than the determined value for the two lignin systems herein and is likely the reason for the lower H_2_S removal capacity of silica. 

The highest removal capacity was determined for lignin precipitated from solution with adsorbed copper at the level of 2 mg H_2_S/g of lignin, which is multiple times lower compared to, e.g., activated carbon [51]. In the current work, our main goal was to study the reactivity and no effort has been made to increase the porosity and surface area of the material. Therefore, it was not expected that the non-modified kraft lignin could reach the performance of specifically activated carbons with very high surface areas. However, certain H_2_S removal technologies for gas purification, such as amine absorption and biological purification, struggle to completely remove H_2_S [5], while implementation in national grids requires H_2_S concentrations under 5 ppm and applications in fuel cells can require levels below 0.1 ppm [3,52]. This suggests that if the processing conditions and the price levels are convenient, one can potentially utilize the lignin-Cu adsorbents as an additional post-treatment purification system for H_2_S removal for, e.g., fuel cell applications or biogas upgrading that require particularly low levels of H_2_S. This would mean that lignin-Cu would be used for the treatment of pretreated relatively clean gas (with respect to co-contaminants). Regarding processing, solid adsorbents are easy to handle, but in order to use lignin as a sole powder adsorbent (and without carbonization), the porosity and surface area of the particles would need to be increased without substantially reducing the concentration of functional groups. Examples of such methods could be related to the works of Li et al. [53] or Cantu et al. [54]. Specific modification of lignin can further increase the metal ion capacity, where values as high as 400 mg of Cu(II) per gram of lignin-based adsorbent have already been achieved via the carboxymethylation of lignin [55], where the H_2_S removal capacity could be enhanced in that manner. Increasing the capacity would also make studying the copper recovery relevant, which regarding lignin-based adsorbents, currently relies on desorption at low pH levels [56,57]. 

Biogas is typically saturated with water that is removed during the upgrading process [58]. H_2_S removal technologies, such as iron oxides and activated carbons, require a certain level of humidity in order to achieve high removal capacities, but too high a water content can negatively influence the process [58,59]. The humidity is believed to enable H_2_S dissociation on the surface of the carbon, which promotes redox reactions with dissociatively adsorbed oxygen [51]. For these reasons and to obtain a better understanding of our system, it was important to study whether the humidity level influenced the adsorption capacity and the H_2_S removal mechanisms. In our experiments, by saturating the air with humidity, the adsorption capacity decreased by 10–40% (Table 1) for all the samples; this issue is further discussed in the following section.

### 3.3. X-ray Photoelectron Spectroscopy

After the H_2_S adsorption, the silica samples and silica/kraft lignin samples with a higher capacity were collected and studied using XPS in order to understand the mechanisms and determine whether copper (II) sulfide was formed or a reaction was occurring on the adsorbent material. The determined peak positions are summarized in Table 2, while Figure 3 shows the copper and sulfur 2p XPS spectra for the silica sample.

The two Cu 2p_3/2_ peaks of 932.4 eV and 933.5 eV were both assigned to monovalent Cu(I) with a formal oxidation state +1 (Cu_2−x_S) [60]. The binding energy of 933.5 eV is also characteristic of Cu(II) [61], but the absence of a shake-up satellite peak (at approximately 944 eV) [62] excluded this possibility. The first peak at 932.4 eV was 0.2–0.3 eV lower than the value expected for CuS and indicated a S-rich nonstoichiometric environment. This was confirmed by the S 2p_3/2_ peak at 162.7–162.9 eV, which is higher than the typical S 2p_3/2_ peaks for Cu_2_S that are expected at 161.7–161.9 eV [63]. The presence of the two peaks in Figure 3b arose from a spin–orbit splitting between the S 2p_1/2_ and S2p_3/2_ states. Finally, the Cu (LMM) peak at 568.7–568.9 eV confirmed that the formal oxidation state of copper was Cu(I), since for Cu (0), the peak is expected at 568.1 eV [64]. To summarize, on the porous silica sample after H_2_S adsorption, the formal oxidative states could be confirmed as +1 for copper and −1 for sulfur.

For the sample that was a silica/lignin mixture, the situation was found to be somewhat different (Figure 4). The position of the S2p_3/2_ 163.8–164.0 peak could be ascribed to elemental S^0^ [65]. Another possibility does exist, as unpassivated sulfur atoms in a Cu_2−x_S compound were found at a similar position [60]. Regarding copper, similar observations were made as that of the previous sample. Cu(II) and Cu^0^ oxidation states were excluded in the same manner. The Cu 2p_3/2_ peaks at 933.1–933.6 eV were assigned to Cu(I) in a Cu-rich environment, which could explain the high binding energy compared to Cu_2_S, while the Cu (LMM) at 571.6 eV was higher than expected for Cu(I), as has been observed before [64]. It was interesting to see that while the situation with copper did not change substantially, in the presence of lignin, the dominant form of sulfur was its elemental form. It seems that in the presence of lignin, the electrochemical potential of the reaction changed, allowing for a higher oxidative state of sulfur to emerge. It was previously found that the adsorption of transition metal ions, such as iron, on polymeric lignin model compounds can be responsible for the oxidation of phenolic groups to species like semiquinonic radicals [66]. The occurrence of these and other types of radicals, for example, hydroxyl radicals, can explain the formation of elemental sulfur. 

When we observe the peak positions for all the samples in Table 2, it is clear that the mechanisms involved were the same, irrespective of the relative air humidity. It appears that low relative humidity is sufficient for effective changes in oxidative states of copper and sulfur, likely by enabling the dissociation of H_2_S on the surface of the adsorbent material. As the mechanisms are the same, a lower H_2_S adsorption capacity with 100% relative humidity of air (Table 1) is possibly explained by water molecules competing for the same sites on the adsorbent material. Another possibility could be the development of a similar situation observed with iron oxides, where water is necessary for the redox process, but a high water content leads to a decrease in available adsorption sites since the oxide surface area decreases due to the adsorbent material sticking together [58].

## 4. Conclusions

This study has shown that during H_2_S adsorption in the presence of kraft lignin, elemental sulfur was formed, while the oxidation state of copper changed to +1. By first solubilizing lignin, it was possible to increase the copper adsorption by several times compared to the adsorption on already-formed lignin particles, and thus, improve the H_2_S removal capacity. The observed H_2_S removal capacity of 2 mg/g of kraft lignin was low and it is clear that to consider the utilization of lignin from this perspective, further studies should be done. For instance, introducing porosity into lignin particles, along with increasing the adsorption of transition metal ions by specific chemical modifications, would positively influence the removal capacity.

## Figures and Tables

**Figure 1 molecules-25-05577-f001:**
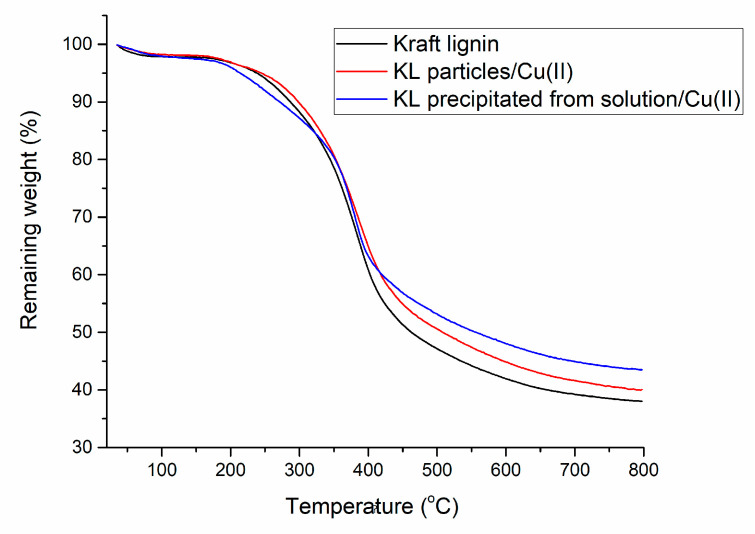
Thermogravimetric analysis for kraft lignin (KL) and the samples with adsorbed copper.

**Figure 2 molecules-25-05577-f002:**
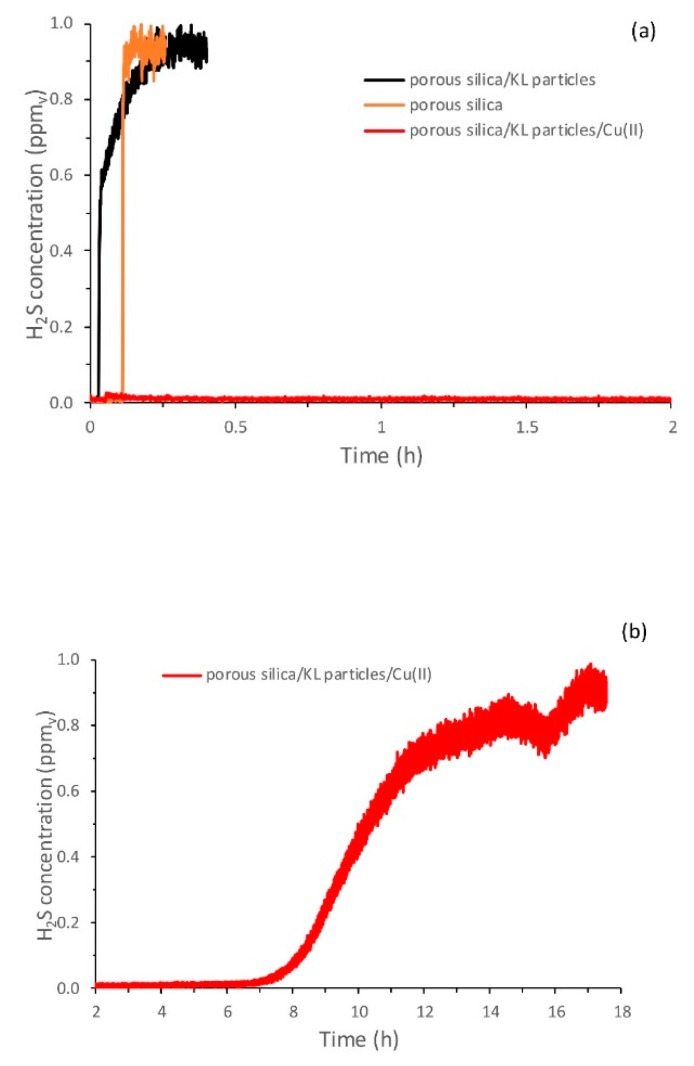
(**a**) Adsorption curves of H_2_S for the silica particles (orange line) and a blend of silica and kraft lignin particles with (red line) and without (black line) adsorbed copper. (**b**) Continuation of the experiment for the blend of particles with adsorbed copper (red line).

**Figure 3 molecules-25-05577-f003:**
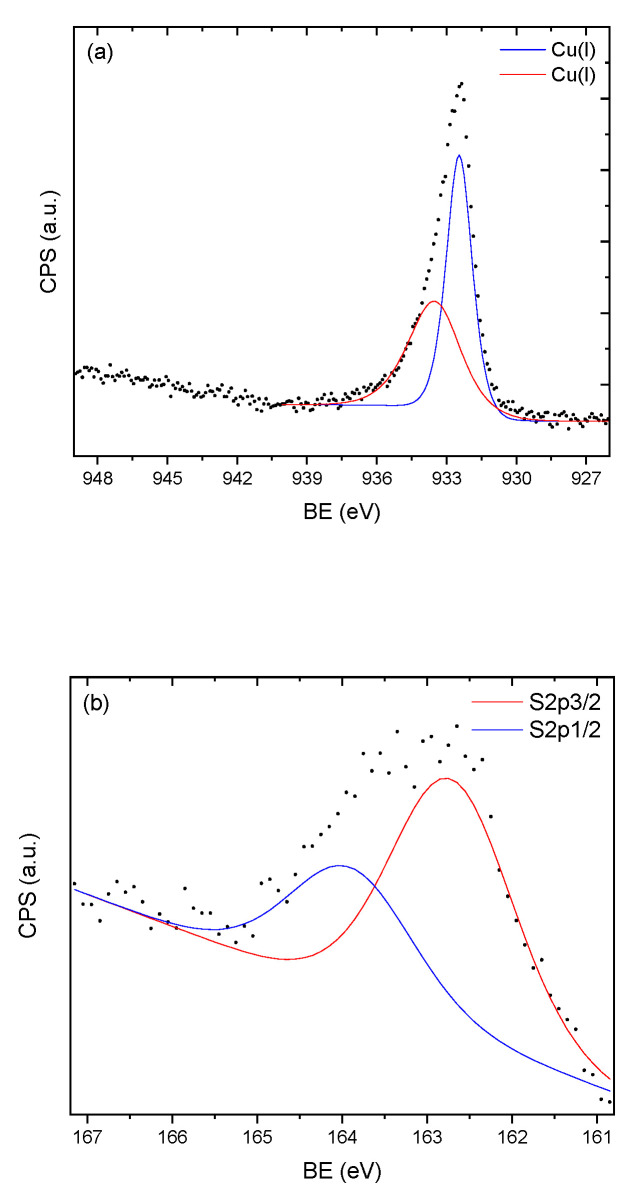
The (**a**) copper 2p_3/2_ and (**b**) sulfur 2p_3/2,1/2_ X-ray photoelectron spectra of the silica sample after H_2_S adsorption. CPS: Counts per second, BE: Binding Energy.

**Figure 4 molecules-25-05577-f004:**
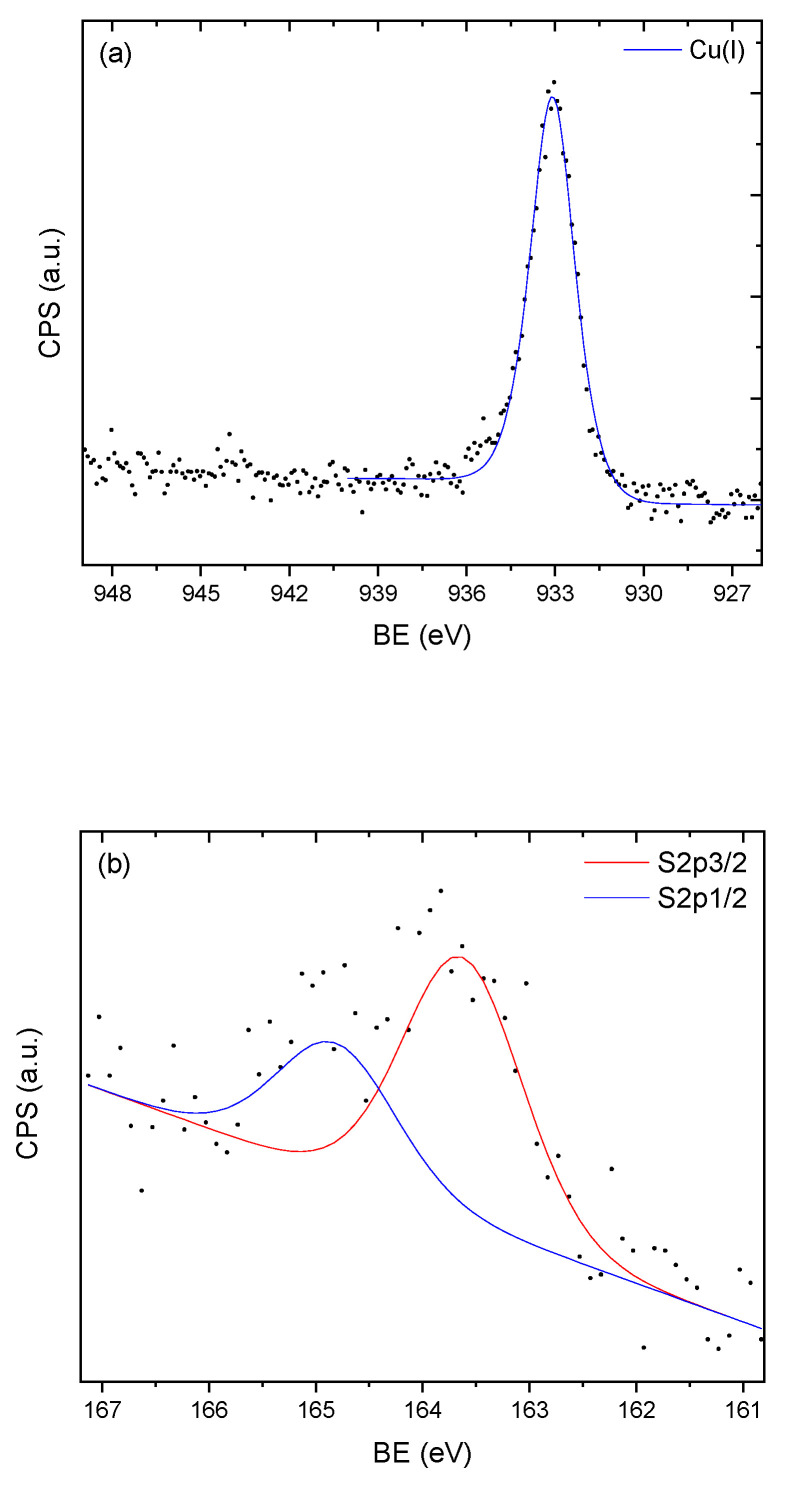
The (**a**) copper *2p_3/2_* and (**b**) sulfur *2p_3/2,1/2_* X-ray photoelectron spectra of porous silica/KL precipitated from the solution/Cu(II) sample after H_2_S adsorption.

**Table 1 molecules-25-05577-t001:** H_2_S adsorption capacities in milligrams per gram of adsorbent material for copper-adsorbed silica, copper-adsorbed kraft lignin particles, and copper-adsorbed kraft lignin obtained via precipitation from a solution, which was tested at low and high relative humidities.

Sample	H_2_S Adsorption Capacity (mg/g)	
	Low Relative Humidity (14% RH)	High Relative Humidity (90–100% RH)
Porous silica/Cu(II)	0.62 ± 0.02	0.38 ± 0.05
KL particles/Cu(II)	0.90 ± 0.39	0.52 ± 0.11
KL precipitated from solution/Cu(II)	2.05 ± 0.40	1.80 ± 0.60

**Table 2 molecules-25-05577-t002:** X-ray photoelectron spectroscopy data for adsorbent materials after H_2_S removal.

Sample	Cu 2p_3/2_	S 2p_3/2_	Cu (LMM *)
Porous silica/Cu(II)	932.4, 933.5	162.7	568.4, 571.7
Porous silica/Cu(II) (adsorption at 90–100% RH)	932.4, 933.5	162.9	568.9, 572.3
Porous silica/KL precipitated from solution/Cu(II)	933.1	163.8	571.6
Porous silica/KL precipitated from solution/Cu(II) (adsorption at 90–100% RH)	933.6	164.0	571.6

* LMM = Auger transition.

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
