# Peer review of "Copper Adsorption on Lignin for the Removal of Hydrogen Sulfide"

_molecules, 2020, doi:10.3390/molecules25235577_

Round 1
Reviewer 1 Report
I still stand by my view that the work should include real systems instead of solely model systems. You have done work on model systems. So what? What are the implications?
As I have previously stated, other components present in real systems could significantly alter the conclusions reached in this study.
Author Response
The authors would like to express their gratitude for your suggestions and helpful comments.
Please find the answer to your comments and suggestions on the attached document.

Reviewer 2 Report
The authors have made some minor changes to the manuscript but I do not see my previous comments have been fully addressed in this round of revision.
- The novelty of this work is still not clear enough. Biopolymer-supported metal-based adsorbents has been widely reported. These previously studies should be mentioned in the Introduction section. Then, what’s new in this work and why this work is significant should be clearly stated.
- The discussion of the results in many places should still be enhanced. For examples, KL precipitated from solution/Cu(II) is shown to have greater adsorption capacity (Table 1). But the reason for this enhancement is not discussed in detail. If KL acted just as a support and have some interaction with Cu, how this could enhance the capability of Cu?
Author Response

(The authors gave the same response as above.)

Reviewer 3 Report
The current paper describes an innovative way of valorizing lignin by adsorbing copper onto the lignin surface and further investigating the potential of the lignin-copper system to remove gaseous hydrogen sulfide. Even though the adsorption potential of the developed material was low when compared to activated carbon, the authors provide new insights on the mechanism of adsorption of H2S onto copper-lignin system, which in my opinion is a valuable information for future researchers who would work on this field. As a whole, the paper is well written, organized, coherent, and easy to understand. The introduction provides relevant references related to the study and sufficiently rationalizes the significance and novelty of the paper. The experimental design seems to be well planned, systematic, and evaluates the important parameters/factors needed to reach the objectives of the study. The results and discussion were succinctly described and explained, and relevant references were cited to support and/or compare their findings. The conclusion was well-justified and does not deviate from their findings. With these, I recommend the paper for publication.
Minor comments:
Copper was incorrectly written in line 62.
What is the sulfur content of the lignin that was used? Would that sulfur have an effect on the XPS analysis?
Author Response

(The authors gave the same response as above.)

Reviewer 4 Report
This manuscript reports the copper adsorption on lignin for removal of hydrogen sulfide. Here are several comments below.
1) In the introduction part, author described “~ is mostly driven by a large fraction of functional groups that are available processes of lignin isolation ~”. However, in order to clarify, it is recommended that author further describes the detailed explanation including the related more references for “a large fraction of functional groups” and “available processes”, which make the reader ambiguous.
2) It could be beneficial to add the possible adsorption mechanism (as a schematic illustration) for the H2S adsorption on lignin structure used in this work.
3) Regarding the solubilizing lignin, why the amount of exposed functional groups are increased, which was well not explained in the manuscript? Why total adsorption sites were increased? The detailed explanation should be kindly provided.
4) What is the size of lignin particles?
5) Importantly, how is the competitive adsorption behaviors against various ions, for instance, Cu2+, K+, Ca2+, Fe2+/3+, etc. Is it selective adsorption of Cu2+? The detailed explanation should be further provided.
6) Moreover, H2S removal of kraft lignin is reversible? Does it can be reused?
Author Response

(The authors gave the same response as above.)

Round 2
Reviewer 1 Report
Acceptable for publication.
Reviewer 2 Report
I do not have further comments.
Reviewer 4 Report
It can be accepted.